# A structured additive modeling of diabetes and hypertension in Northeast India

**Strong P. Marbaniang**[1,2]*, **Holendro Singh Chungkham**[3,4], **Hemkhothang Lhungdim**[1]

1 Department of Public Health & Mortality studies, International Institute for Population Sciences, Mumbai, India, 2 Department of Statistics, Sankardev College, Shillong, Meghalaya, India, 3 Indian Statistical Institute, North-East Centre, Tezpur, Assam, India, 4 Stress Research Institute, Stockholm University, Stockholm, Sweden

☯ These authors contributed equally to this work.
* strongmarbaniang@yahoo.com

**Data Availability Statement:** The data can be found from the following link: https://dhsprogram.com/data/dataset/India_Standard-DHS_2015.cfm?flag=1.

## Abstract

### Background

Multiple factors are associated with the risk of diabetes and hypertension. In India, they vary widely even from one district to another. Therefore, strategies for controlling diabetes and hypertension should appropriately address local risk factors and take into account the specific causes of the prevalence of diabetes and hypertension at sub-population levels and in specific settings. This paper examines the demographic and socioeconomic risk factors as well as the spatial disparity of diabetes and hypertension among adults aged 15–49 years in Northeast India.

### Methods

The study used data from the Indian Demographic Health Survey, which was conducted across the country between 2015 and 2016. All men and women between the ages of 15 and 49 years were tested for diabetes and hypertension as part of the survey. A Bayesian geo-additive model was used to determine the risk factors of diabetes and hypertension.

### Results

The prevalence rates of diabetes and hypertension in Northeast India were, respectively, 6.38% and 16.21%. The prevalence was higher among males, urban residents, and those who were widowed/divorced/separated. The functional relationship between household wealth index and diabetes and hypertension was found to be an inverted U-shape. As the household wealth status increased, its effect on diabetes also increased. However, interestingly, the inverse was observed in the case of hypertension, that is, as the household wealth status increased, its effect on hypertension decreased. The unstructured spatial variation in diabetes was mainly due to the unobserved risk factors present within a district that were not related to the nearby districts, while for hypertension, the structured spatial variation was due to the unobserved factors that were related to the nearby districts.

**Funding:** The author(s) received no specific funding for this work.

**Competing interests:** The authors have declared that no competing interests exist.

## Conclusion

Diabetes and hypertension control measures should consider both local and non-local factors that contribute to the spatial heterogeneity. More importance should be given to efforts aimed at evaluating district-specific factors in the prevalence of diabetes within a region.

## Introduction

Diabetes and hypertension are major global health concerns. They impose a heavy burden on the public healthcare sector and affect socioeconomic development [1,2]. Statistics from the International Diabetes Federation (IDF) and the World Health Organization (WHO) show that about 463 million adults were living with diabetes in 2019 [3] and 1.13 billion with hypertension in 2015 [4]. According to estimates from 2019, India had the second-highest number (77 million) of diabetic people in the world, and the number is expected to increase to 134 million by 2045 [3]. Evidence from a study based on the Demographic Health Survey shows that 11.3% of Indians aged 15–54 years have diabetes, with the prevalence being higher among men (13.8%) than women (8.8%) [5].

Northeast India, which is located in the Northeastern part of India, is composed of eight small states, namely Assam, Arunachal Pradesh, Manipur, Meghalaya, Mizoram, Nagaland, Sikkim, and Tripura. The region is mostly inhabited by tribal communities belonging to different ethnic groups [6]. Geographically, the region is mostly hilly, which acts as a major hurdle to transportation and communication, affecting the access to and the proper functioning of healthcare facilities in the rainy season [7]. Despite low per capita income, the prevalence of hypertension in the region is much higher than in states that are more socioeconomically developed and have much higher per capita incomes [8]. According to the Indian Demographic Health Survey (2019–20) report, the prevalence of diabetes and hypertension in Northeast India is higher among men than women. As per the survey report, 15.6 percent of men have diabetes as compared to 12.7 percent of women, and 27.6 percent of men have hypertension as compared to 22.3 percent of women [9].

The epidemiology of diabetes and hypertension reveals multiple risk factors. Previous studies have shown that socioeconomic factors–such as low levels of education, high household economic status, and demographic factors like age and sex–increase the risk of diabetes and hypertension [10,11]. Lifestyle behaviours like smoking, alcohol consumption [11–13], low physical activity [14], and dietary habits [15,16] also significantly influence the risk of diabetes and hypertension. Individuals in the same geographical area usually have common beliefs and culture, which may lead to similar levels of exposure to diseases, including diabetes and hypertension [17–19]. Hence, countries with a diverse culture and wide differences in dietary habits are likely to have large variations in the prevalence of diabetes and hypertension based on their geographical location [20,21].

Despite the diversity in dietary habits and cultural practices, studies on diabetes and hypertension in Northeast India have not, to our knowledge, investigated the geographical heterogeneity in the causes of diabetes and hypertension [22,23]. According to Koissi et al., overlooking the effects of heterogeneity in the statistical model may lead to biased parameter estimates [24]. It is important to note here that geographical heterogeneity can be an effect of unobserved factors that may be mostly contextual. Geographical differences in the causes of diabetes and hypertension can be explained by large-scale variability in environmental factors like availability of green spaces in a catchment area of 1 km radius around the residential

location [25], level of urbanization and westernization [26], differences in dietary patterns [20,21], level of poverty, and access to medical facilities [27]. Studies have shown that obesity, which leads to diabetes and hypertension, is associated with the availability of green spaces or parks [25,28]. A study by Haynes-Maslow et al. showed that an increase in the number of fast-food restaurants in a county is associated with increasing prevalence of diabetes in that particular county [29]. Several studies from India and abroad [18,20,21] have considered geographical heterogeneity while modeling diabetes and hypertension; however they have overlooked the non-linear effects of continuous variables (that is, using the bivariate spline approach) while modeling the geographical heterogeneity.

This paper contributes to the understanding of spatial variations in diabetes and hypertension in Northeast India by using the Bayesian spatial mixed model approach, which is based on the Markov Chain Monte Carlo (MCMC) simulation technique. To the best of the authors' knowledge, this study is the first to map diabetes and hypertension in Northeast India in terms of the spatial effect. The map is likely to have significant implications for our understanding of how diabetes and hypertension are spatially distributed and will help health promotion programmes allocate the resources equitably and efficiently.

## Material and methods

### Study area and data

The focus of the study was Northeast India. Data used in the analysis was drawn from the nationally representative Indian Demographic Health Survey (IDHS), also known as the National Family Health Survey (NFHS-4) which was conducted across the country between 2015 and 2016. The Indian Demographic Health Survey (IDHS) was conducted by the International Institute for Population Sciences (IIPS), Mumbai, a nodal agency appointed by the Ministry of Health and Family Welfare, Government of India [30]. After completing the registration for getting the approval to download the dataset, the data can be downloaded from the DHS website [31]. Since this study used publicly available secondary data and de-identified the respondents, the institutional review board (IRB) exempted it from seeking approval.

The survey employed a two-stage stratified sampling design. In the first stage, primary sampling units (PSUs) were selected based on probability proportional to population size. In rural areas, villages were the PSUs, while in urban areas, census enumeration blocks formed the PSUs. In every selected rural and urban PSU, a complete household mapping and listing was conducted prior to the main survey. Among the selected PSUs, those having at least 300 households were divided into segments of 100–150 households. In NFHS-4, a cluster is either a PSU or a segment of a PSU. In the second stage, 22 households were selected from every selected urban and rural cluster using the systematic random sampling method. From each selected household, information was sought from women aged 15–49 years and men aged 15–54 years [30]. The study excluded Sikkim because its boundary is not connected to the map of Northeast India and including it would have made estimating the spatial effects difficult (Fig 1). The shapefile map used in this study was downloaded from the website of GADM and can be used under the Creative Commons Attribution License (CCAL), CC BY 4.0 [32].

### Sampling

The sample for this study comprised 112,062 respondents (98,702 females and 13,360 males) aged 15–49 years. Males comprised only 12 percent of the total sample size because the survey had collected information from males from only 15 percent of the sampled households. A total of 6,878 respondents had diabetes, while 17,677 respondents had hypertension. The study covered 82 districts, whose breakup by states is given in the supporting file S1 Table.

## Location Map

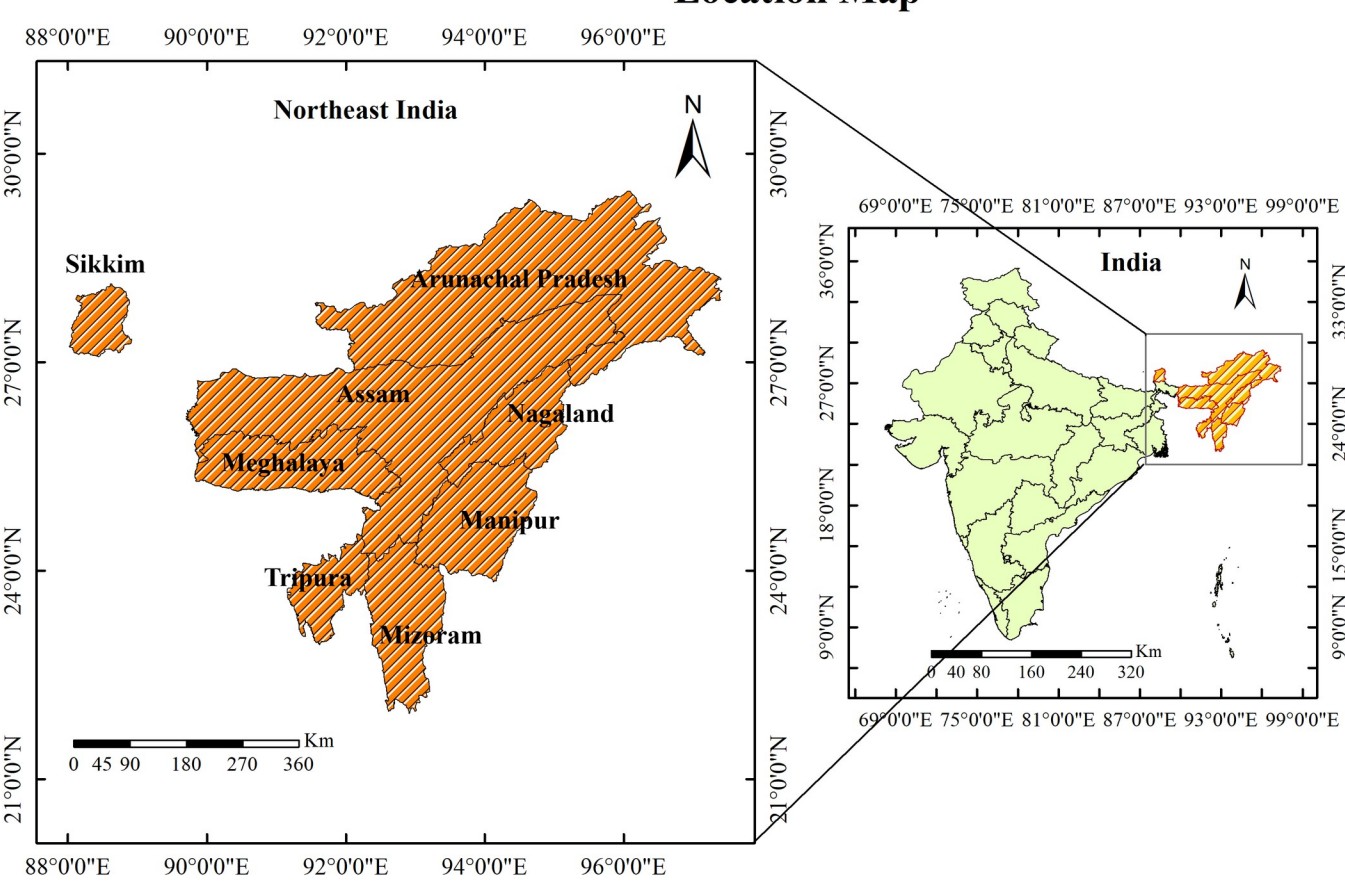

**Fig 1. Map showing the location of the study area.**

### Operational definitions

**Diabetes.** A FreeStyle Optium H Glucometer device was used to measure the blood glucose. The device uses a small drop of blood drawn from the fingertips to measure the blood glucose level. The blood sample was drawn only once at a random time during the day irrespective of when the respondent last ate. Usually, the presence or absence of diabetes in an individual is determined on the basis of fasting blood glucose level. However, NFHS-4 measured the random blood glucose level. A respondent is considered to have diabetes if the random blood glucose level is >140mg/dl.

**Hypertension.** Blood pressure was measured with an Omron Hem 7203 blood pressure monitor. Three blood pressure readings were taken in all, with an interval of 5 minutes between the readings. The first reading was discarded and the average of the last two readings was calculated. A respondent was classified as hypertensive if the average systolic blood pressure was $\geq$ 140 mmHg, or if the average diastolic blood pressure $\geq$ 90 mmHg, or if the person was taking antihypertensive medication to lower blood pressure at the time of the survey [30].

### Dependent variables

The outcome variables were diabetes and hypertension status of a respondent. The values were binary, with 1 implying "Yes" (meaning presence of diabetes or hypertension) and 0 implying "No" (meaning absence).

## Explanatory variables

The choice of the explanatory variables was guided by the existing literature. The demographic variables considered in the study were age and sex of the respondents. The socioeconomic variables included the respondents' caste, marital status, level of education, place of residence, and household wealth. The variables for lifestyle behaviors included cigarette smoking and tobacco and alcohol consumption. To capture the effects of dietary habits on chronic diseases, we included foods consumed by the respondents and categorized them as milk, pulses, vegetables, fish, fruits, eggs, chicken, aerated drinks, and fried food. The fixed effects are compared according to the effect-coding given in Table 1.

The continuous explanatory variables for the study were age of the respondents (in years), body mass index (kg/m$^2$), and wealth index score.

## Statistical analysis

A multiple logistic regression was applied to select the potential covariates of diabetes and hypertension prior to the spatial analysis. To allow for more potential covariates for the spatial analysis, a significance level of 20% was set for the selection of the potential covariates. They are listed in Table 1.

The traditional linear regression model has the limitation of not being able to incorporate spatial and non-linear effects more flexibly in the model. For a study like ours, where the primary objective was to explore unobserved heterogeneity in the structured and unstructured spatial effects, geo-additive models were better suited than the linear regression models. Therefore, the data were fitted using geo-additive logistic regression models to understand the fixed as well as the spatial effects for diabetes and hypertension (the term chronic disease was used in place of diabetes and hypertension). The respondents' status of chronic disease was a binary outcome; it was distributed as *Bernoulli* ($p_{ij}$) where $p_{ij}$ was the probability that respondent $j$ in district $i$ had a chronic disease. The district of the respondent was labelled as $s_i \epsilon$ (1, 2, 3. . ..,82), where the label matched the labels on the map. The spatial effect of district $s_i$, in which the respondent resided, was given by $f_{spatial}(s_i)$. The spatial effect comprised two parts: a spatially correlated (or structured) effect and an uncorrelated (or unstructured) effect. Thus,

$$f_{spatial}(s_i) = f_{structured}(s_i) + f_{unstructured}(s_i)$$

The following models were fitted to estimate the fixed and spatial effects.

M0: logit $(p_{ij}) = z_i^T \beta$

M1: logit $(p_{ij}) = z_i^T \beta + f_1(u_{i1}) + f_2(u_{i2}) + f_3(u_{i3}) + \ldots \ldots + f_p(u_{ip})$

M2: logit $(p_{ij}) = z_i^T \beta + f_{structured}(s_i) + f_{unstructured}(s_i)$

M3: logit $(p_{ij}) = z_i^T \beta + f_1(u_{i1}) + f_2(u_{i2}) + f_3(u_{i3}) + \ldots \ldots + f_p(u_{ip}) + f_{structured}(s_i)$

M4: logit $(p_{ij}) = z_i^T \beta + f_1(u_{i1}) + f_2(u_{i2}) + f_3(u_{i3}) + \ldots \ldots + f_p(u_{ip}) + f_{unstructured}(s_i)$

M5: logit $(p_{ij}) = z_i^T \beta + f_1(u_{i1}) + f_2(u_{i2}) + f_3(u_{i3}) + \ldots \ldots + f_p(u_{ip}) + f_{structured}(s_i) + f_{unstructured}(s_i)$

In model M0, all the categorical and continuous variables were considered as fixed effects, and $\beta$ was the parameter in the vector form. In model M1, categorical variables were treated as fixed effects, while continuous variables were modelled as a non-parametric smooth function $f_j$s. In model M2, all the covariates were modelled as fixed covariates, and the district of the respondent was modelled as a spatial effect. Model M3 was a combination of M1 and M2 in which the smooth function $f_j$s was assigned with Bayesian P-spline priors and the spatial effect

**Table 1. Prevalence of diabetes and hypertension among adults aged 15–49 years by fixed covariates with effect coding used in the model).**

| Variables | Diabetes | | Hypertension | | Effect Coding |
|---|---|---|---|---|---|
| | (%) | P* | (%) | P* | |
| **Sex** | | | | | |
| Female | 6.16 | 0.000 | 15.59 | 0.000 | -1@ |
| Male | 8.03 | | 20.84 | | 1 |
| **Residence** | | | | | |
| Rural | 5.88 | 0.000 | 15.98 | 0.000 | -1@ |
| Urban | 7.73 | | 16.85 | | 1 |
| **Current marital status** | | | | | |
| Never married | 4.00 | 0.000 | 8.43 | 0.000 | -1@ |
| Married | 7.18 | | 19.15 | | 1 |
| Widowed/Divorced/Separated | 9.41 | | 22.24 | | 2 |
| **Caste** | | | | | |
| Scheduled tribe | 6.40 | 0.639 | 15.74 | 0.000 | -1@ |
| Scheduled caste | 6.16 | | 16.89 | | 1 |
| Others | 6.45 | | 16.63 | | 2 |
| **Level of education** | | | | | |
| Illiterate | 7.06 | 0.000 | 22.27 | 0.000 | -1@ |
| Primary | 7.21 | | 17.55 | | 1 |
| Secondary | 5.84 | | 14.20 | | 2 |
| Higher secondary | 7.13 | | 15.50 | | 3 |
| **Consume milk** | | | | | |
| No | 7.01 | 0.000 | 16.38 | 0.529 | -1@ |
| Yes | 6.27 | | 16.18 | | 1 |
| **Consume pulses** | | | | | |
| No | 5.71 | 0.400 | 20.87 | 0.000 | -1@ |
| Yes | 6.38 | | 16.17 | | 1 |
| **Consume vegetables** | | | | | |
| No | 4.21 | 0.152 | 18.42 | 0.328 | -1@ |
| Yes | 6.38 | | 16.21 | | 1 |
| **Eat fruits** | | | | | |
| No | 7.15 | 0.189 | 20.52 | 0.000 | -1@ |
| Yes | 6.36 | | 16.14 | | 1 |
| **Consume egg** | | | | | |
| No | 8.16 | 0.000 | 20.13 | 0.000 | -1@ |
| Yes | 6.32 | | 16.10 | | 1 |
| **Eat fish** | | | | | |
| No | 7.26 | 0.069 | 15.74 | 0.518 | -1@ |
| Yes | 6.36 | | 16.22 | | 1 |
| **Eat chicken** | | | | | |
| No | 8.26 | 0.000 | 19.35 | 0.000 | -1@ |
| Yes | 6.33 | | 16.13 | | 1 |
| **Eat fried food** | | | | | |
| No | 6.73 | 0.349 | 21.23 | 0.000 | -1@ |
| Yes | 6.36 | | 16.01 | | 1 |
| **Take aerated drinks** | | | | | |
| No | 6.88 | 0.000 | 17.31 | 0.000 | -1@ |

(*Continued*)

**Table 1.** (Continued)

| Variables | Diabetes | | Hypertension | | Effect Coding |
|---|---|---|---|---|---|
| | (%) | P* | (%) | P* | |
| Yes | 6.23 | | 15.90 | | 1 |
| **Consume alcohol** | | | | | |
| No | 6.20 | 0.000 | 15.16 | 0.000 | -1@ |
| Yes | 7.42 | | 22.51 | | 1 |
| **Currently smoke cigarettes** | | | | | |
| No | 6.24 | 0.000 | 16.11 | 0.000 | -1@ |
| Yes | 8.56 | | 17.91 | | 1 |
| **Consume tobacco** | | | | | |
| No | 6.28 | 0.000 | 16.24 | 0.194 | -1@ |
| Yes | 9.27 | | 15.43 | | 1 |

@: Reference category

*: p-value of chi-square test of independence.

$f_{structured}(s_i)$ with Markov random field priors [33,34]. We considered a fourth model, M4, which was again a combination of M1 and M2, where the smooth function $f_j s$ was assigned with Bayesian P-spline priors and the spatial effect as $f_{unstructured}(s_i)$. We considered a fifth and the final model, M5, which was a combination of M3 and M4, where both structured and unstructured spatial effects were included. The spatial effects represented the effects of the unobserved covariates that were not incorporated in the model and accounted for spatial autocorrelation.

The structured spatial effect $f_{structured}(s_i)$ accounted for the spatial variation due to the unobserved influences that arose due to the assumption that the nearby districts were likely to be correlated with respect to their outcomes. However, in the case of the unstructured spatial effect, $f_{unstructured}(s_i)$, the spatial variation was due to the unobserved influences that were present locally, that is, within a district. Markov random field (MRF) priors were specified for the structured spatial effect. Two districts were defined as neighbors if they shared a common boundary. The conditional mean of $f_{structured}(s_i)$ was an average of the evaluations of $f_{structured}(s_i)$ of other neighboring districts. In the same way, $i.i.d$ Gaussian priors were assigned for the unstructured spatial effects $f_{unstructured}(s_i)$.

A fully Bayesian integrated approach, based on the Markov Chain Monte Carlo (MCMC) simulation, was used to estimate the model parameters. The estimated prior odds ratio (OR) could be interpreted as the odds ratio from the logistic regression. The model was fitted in R using the freely available package *bamlss* [35]. For the analysis, we used a total of 40,000 MCMC iterations and 10,000 burns in the sample. Convergence checks of the models were based on autocorrelation and the sampling paths. Finally, all the models used in the analysis were compared using the Deviance Information Criterion (DIC) values [36]; the model with the smallest DIC values was preferred for estimating the parameters. $DIC$ is defined as $DIC = \bar{D} + p_D$, where $\bar{D}$ is the posterior mean of the model deviance, which is a measure of goodness of fit, and $p_D$ is the effective number of parameters, which indicates the complexity of the model and penalizes over-fitting.

## Results

### Descriptive statistics

Table 1 shows the prevalence of diabetes and hypertension across the categorical covariates. It is evident from the results that males, urban residents, and widowed, divorced or separated

individuals had a higher prevalence of diabetes and hypertension. There was a significant gender difference in the prevalence of diabetes and hypertension. This difference was also seen for place of residence, marital status, and educational level of the respondents.

The prevalence of diabetes and hypertension was the highest among respondents who were widowed, divorced or separated. The results also show that the prevalence of diabetes was lower among respondents who consumed milk than those who did not. However, this association was not significant for the prevalence of hypertension. Consuming fruits and fried foods showed a positive impact in reducing the prevalence of hypertension. Unhealthy lifestyle behaviors, such as cigarette smoking and drinking alcohol, were significantly associated with a high prevalence of diabetes and hypertension. Since all the categorical variables listed in Table 1 showed a significant association with diabetes and hypertension at 20% level of significance in the preliminary analysis, they were all included in the spatial logistic regression model (Tables 3 and 4).

### Empirical bayesian results

**Model selection.** The selection of a better model is based on DIC and deviance values. A model with the smallest DIC and deviance values is considered the best model. It can be seen from Table 2 that model M5 had the smallest DIC and deviance values for both diabetes and hypertension. Models with differences in DIC values less than 3 cannot be differentiated, while those with values between 3 and 7 can be weakly differentiated [37]. Taking all of these criteria into account, this study based the interpretation of the results of the analysis on model M5, the geo-additive model with both structured and unstructured spatial effects.

**Fixed effects.** In model M5, the effects of the categorical covariates were assumed to be fixed and were estimated jointly with the continuous and spatial covariates. The posterior means and the corresponding 97.5% credible intervals of the fixed effects parameters are shown in Table 3. The fixed effects covariates which were significant to diabetes were sex, current marital status, level of education, and consumption of tobacco. The fixed effect coefficient for males was positive, which indicates that being male increased the risk of diabetes as

**Table 2. Comparison of models based on deviance information criterion (DIC).**

| Diabetes | Model Fit | Deviance ($\bar{D}$) | $p_D$ | DIC | $\Delta^\$ DIC$ |
|---|---|---|---|---|---|
| | M0 | 42763.22 | 9.9387 | 42773.17 | 2941.24 |
| | M1 | 46648.94 | 73.9609 | 46722.91 | 6890.98 |
| | M2 | 40199.76 | 101.2486 | 40301.01 | 469.08 |
| | M3 | 39725.00 | 111.2923 | 39836.30 | 4.37 |
| | M4 | 39722.80 | 112.4136 | 39835.28 | 3.35 |
| | M5 | 3971844 | 113.4988 | 39831.93 | *Reference* |
| Hypertension | Model Fit | Deviance ($\bar{D}$) | $p_D$ | DIC | $\Delta^\$ DIC$ |
| | M0 | 79703.78 | 9.8346 | 79713.62 | 7402.04 |
| | M1 | 88477.04 | 77.5036 | 88554.55 | 16242.97 |
| | M2 | 73197.80 | 105.1395 | 73302.95 | 991.37 |
| | M3 | 72204.20 | 112.3875 | 72316.50 | 4.92 |
| | M4 | 72200.80 | 113.9219 | 72314.78 | 3.20 |
| | M5 | 72327.87 | 115.5265 | 72311.58 | *Reference* |

*M0:Categorical and continuous covariates were treated as fixed effect; M1:Categorical were treated as fixed and continuous as non-linear effect; M2: All covariates were treated as fixed effect, and districts as spatial effect; M3: Combination of M1 and M2 with only structured spatial effect; M4: Combination of M1 and M2 with only unstructured spatial effect; M5: Combination of M3 and M4; $^\$$: Difference of M5 against M0, M1, M2, M3 and M4.*

**Table 3. Posterior estimates of the fixed effects parameters for diabetes in Northeast India.**

| Variables | Mean | SD | 2.5% Quantile | Median | 97.5% Quantile |
|---|---|---|---|---|---|
| **Sex** | | | | | |
| Female@ | | | | | |
| Male | 0.138* | 0.025 | 0.088 | 0.137 | 0.187 |
| **Residence** | | | | | |
| Rural@ | | | | | |
| Urban | 0.028 | 0.017 | -0.006 | 0.028 | 0.064 |
| **Current marital status** | | | | | |
| Never married@ | | | | | |
| Married | -0.104* | 0.025 | -0.151 | -0.105 | -0.056 |
| Widowed/Divorced/Separated | -0.030 | 0.040 | -0.110 | -0.030 | 0.044 |
| **Caste** | | | | | |
| Scheduled tribe@ | | | | | |
| Scheduled caste | -0.030 | 0.034 | -0.095 | -0.028 | 0.033 |
| Others | -0.028 | 0.036 | -0.099 | -0.028 | 0.044 |
| **Level of education** | | | | | |
| Illiterate@ | | | | | |
| Primary | 0.032 | 0.031 | -0.029 | 0.031 | 0.096 |
| Secondary | -0.031 | 0.024 | -0.077 | -0.030 | 0.015 |
| Higher secondary | -0.062* | 0.032 | -0.121 | -0.062 | -0.001 |
| **Consume milk** | | | | | |
| No@ | | | | | |
| Yes | -0.025 | 0.020 | -0.062 | -0.026 | 0.016 |
| **Consume pulses** | | | | | |
| No@ | | | | | |
| Yes | 0.075 | 0.080 | -0.069 | 0.076 | 0.235 |
| **Consume vegetables** | | | | | |
| No@ | | | | | |
| Yes | 0.054 | 0.171 | -0.269 | 0.050 | 0.392 |
| **Eat fruits** | | | | | |
| No@ | | | | | |
| Yes | -0.076 | 0.050 | -0.175 | -0.075 | 0.025 |
| **Consume egg** | | | | | |
| No@ | | | | | |
| Yes | -0.043 | 0.045 | -0.137 | -0.044 | 0.044 |
| **Eat fish** | | | | | |
| No@ | | | | | |
| Yes | -0.078 | 0.054 | -0.181 | -0.077 | 0.024 |
| **Eat chicken** | | | | | |
| No@ | | | | | |
| Yes | -0.007 | 0.050 | -0.108 | -0.008 | 0.088 |
| **Eat fried food** | | | | | |
| No@ | | | | | |
| Yes | 0.005 | 0.038 | -0.069 | 0.005 | 0.082 |
| **Take aerated drinks** | | | | | |
| No@ | | | | | |
| Yes | -0.012 | 0.018 | -0.047 | -0.012 | 0.023 |
| **Consume alcohol** | | | | | |

(*Continued*)

**Table 3.** (Continued)

| Variables | Mean | SD | 2.5% Quantile | Median | 97.5% Quantile |
|---|---|---|---|---|---|
| No@ | | | | | |
| Yes | -0.023 | 0.023 | -0.066 | -0.023 | 0.019 |
| **Smoke cigarettes** | | | | | |
| No@ | | | | | |
| Yes | 0.003 | 0.027 | -0.049 | 0.002 | 0.059 |
| **Consume tobacco** | | | | | |
| No@ | | | | | |
| Yes | -0.038* | 0.016 | -0.069 | -0.038 | -0.006 |

@: Reference category

*: Statistical significance at 5%.

compared to being female. The coefficient for marital status 'married' was negative, which means that married individuals were at a reduced risk of diabetes as compared to never married individuals. Individuals who consumed tobacco were also seen as being at a reduced risk of diabetes.

For hypertension, the posterior means and the corresponding 97.5% credible intervals of the fixed effects parameters are given in Table 4. Urban residence had a positive effect on hypertension, meaning that individuals who lived in urban areas were at an increased risk of hypertension. Individuals who were educated up to the higher secondary (high school) level were found to be less likely to suffer from hypertension than individuals without an education. Consumption of milk showed a negative coefficient, meaning that having milk reduced the risk of hypertension. An interesting finding of our analysis was that individuals who consumed alcohol were at a lower risk of hypertension as compared to those who did not.

**Non-linear effects.** Another important advantage of using the geo-additive model is its ability to incorporate non-linear effects due to continuous covariates. In this study, we incorporated the non-linear effects of body mass index (BMI), wealth index score, and age of the respondents.

Body mass index of individuals had a non-linear effect on diabetes and hypertension (Fig 2). It is evident from Fig 1 that as the BMI increased, its effect on diabetes and hypertension also increased. The risk of diabetes and hypertension was lower at BMI values of 20 to 25; however, the risk increased for BMI values of 50 and more.

Household wealth index scores had a non-linear effect on diabetes and hypertension (Fig 3). The functional relationship between household wealth index and diabetes and hypertension was almost inverted U-shaped. With increasing household wealth status, the effect on diabetes also increased. Interestingly, the reverse was observed in the case of hypertension, that is, as the household wealth status increased, its effect on hypertension decreased.

Age of respondents showed an almost linear relationship with diabetes and hypertension (Fig 4). The effect of age on diabetes and hypertension was the lowest at age 15 years and the maximum at age 49 years.

**Spatial effects.** Figs 5 and 6 present the estimated spatial effects of diabetes and hypertension, with color ranges from blue to maroon indicating low to high risk of diabetes and hypertension. Districts marked in blue had a negative spatial effect and were, therefore, associated with lower odds of diabetes and hypertension. Districts shown in maroon had a positive spatial effect and were, therefore, associated with higher odds of diabetes and hypertension. Spatial

**Table 4. Posterior estimates of the fixed effects parameters for hypertension in Northeast India.**

| Variables | Mean | SD | 2.5% Quantile | Median | 97.5% Quantile |
|---|---|---|---|---|---|
| **Sex** | | | | | |
| Female@ | | | | | |
| Male | 0.159* | 0.017 | 0.125 | 0.160 | 0.192 |
| **Residence** | | | | | |
| Rural@ | | | | | |
| Urban | 0.05* | 0.013 | 0.026 | 0.050 | 0.074 |
| **Current marital status** | | | | | |
| Never married@ | | | | | |
| Married | -0.053* | 0.021 | -0.085 | -0.053 | -0.019 |
| Widowed/Divorced/Separated | 0.006 | 0.024 | -0.051 | 0.007 | 0.063 |
| **Caste** | | | | | |
| Scheduled tribe@ | | | | | |
| Scheduled caste | -0.034 | 0.021 | -0.076 | -0.034 | 0.008 |
| Others | 0.034 | 0.024 | -0.013 | 0.035 | 0.079 |
| **Level of education** | | | | | |
| Illiterate@ | | | | | |
| Primary | 0.009 | 0.021 | -0.032 | 0.008 | 0.051 |
| Secondary | 0.004 | 0.015 | -0.025 | 0.004 | 0.034 |
| Higher secondary | -0.096* | 0.021 | -0.136 | -0.096 | -0.054 |
| **Consume milk** | | | | | |
| No@ | | | | | |
| Yes | -0.036* | 0.015 | -0.065 | -0.036 | -0.007 |
| **Consume pulses** | | | | | |
| No@ | | | | | |
| Yes | -0.074 | 0.048 | -0.065 | -0.076 | 0.023 |
| **Consume vegetables** | | | | | |
| No@ | | | | | |
| Yes | -0.139 | 0.091 | -0.307 | -0.141 | 0.044 |
| **Eat fruits** | | | | | |
| No@ | | | | | |
| Yes | -0.025 | 0.034 | -0.091 | -0.025 | 0.043 |
| **Consume egg** | | | | | |
| No@ | | | | | |
| Yes | -0.071* | 0.032 | -0.132 | -0.072 | -0.010 |
| **Eat fish** | | | | | |
| No@ | | | | | |
| Yes | 0.081* | 0.041 | 0.001 | 0.083 | 0.162 |
| **Eat chicken** | | | | | |
| No@ | | | | | |
| Yes | -0.018 | 0.035 | -0.086 | -0.018 | 0.052 |
| **Eat fried food** | | | | | |
| No@ | | | | | |
| Yes | -0.035 | 0.023 | -0.081 | -0.036 | 0.011 |
| **Take aerated drinks** | | | | | |
| No@ | | | | | |
| Yes | -0.002 | 0.012 | -0.026 | -0.003 | 0.021 |
| **Consume alcohol** | | | | | |

(*Continued*)

**Table 4.** (Continued)

| Variables | Mean | SD | 2.5% Quantile | Median | 97.5% Quantile |
|---|---|---|---|---|---|
| No@ | | | | | |
| Yes | -0.12* | 0.015 | -0.147 | -0.120 | -0.091 |
| **Smoke cigarettes** | | | | | |
| No@ | | | | | |
| Yes | 0.086* | 0.021 | 0.042 | 0.086 | 0.126 |
| **Consume tobacco** | | | | | |
| No@ | | | | | |
| Yes | 0.012 | 0.011 | -0.010 | 0.013 | 0.033 |

@: Reference category

*: Statistical significance at 5%.

effects are surrogates for unknown influences like environmental factors, climate, availability of proper transport, and access to good healthcare facilities.

Fig 5A clearly shows a significant clustering of diabetes in Northeast India, with the risk of diabetes being higher in the districts of Nagaland, Manipur, Mizoram, and Tripura. Districts with low risk of diabetes are in the states of Assam, Arunachal Pradesh, and Meghalaya. However, overall, the whole of Northeast India appears to be less affected by the unstructured spatial effects of diabetes (Fig 5B). The structured spatial effects of diabetes, which ranged from -0.27 to 0.47, were weak in comparison to the unstructured spatial effects, which ranged from -1.51 to 1.71

Fig 6 shows spatial clustering of hypertension. It can be seen that the risk of hypertension was higher in the districts of Assam, Arunachal Pradesh, Nagaland, and Meghalaya and lower in the districts of Manipur, Mizoram, Tripura, and Hills and Barak valley of Assam. In Fig 6B, the unstructured spatial effects of hypertension can be observed in some districts of Arunachal Pradesh (Anjaw, Dibang valley, and West Siang), suggesting that the spatial variation was due to the effect of unmeasured local influences. For hypertension, the structured spatial effects ranged from -0.48 to 0.68, which dominated the unstructured spatial effects (-0.4 to 0.6).

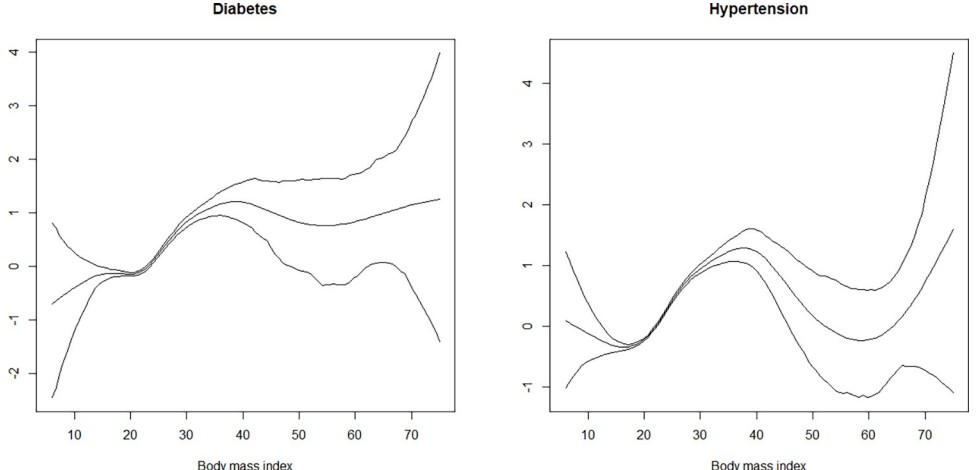

**Fig 2. Non-linear effects of body mass index on the log-odds of diabetes and hypertension (the figure shows posterior means along with the 97.5% credible intervals).**

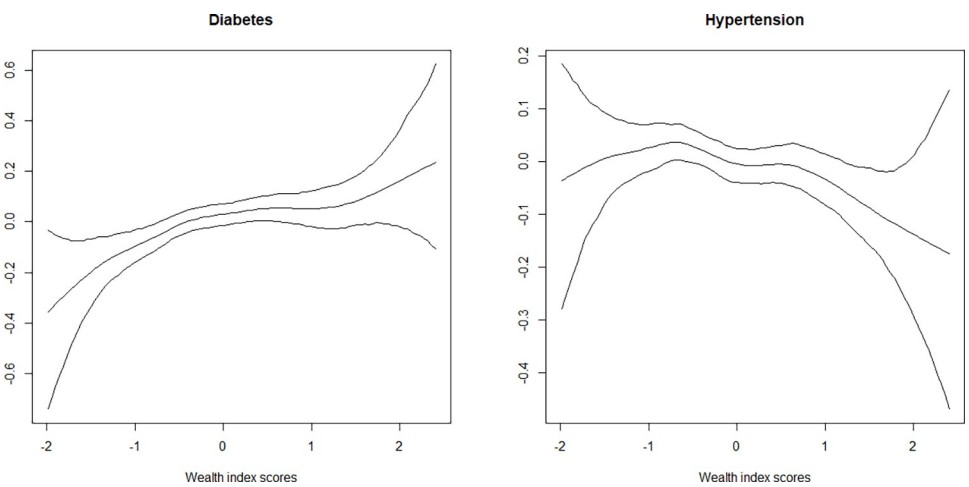

**Fig 3. Non-linear effects of wealth index score on the log-odds of diabetes and hypertension (posterior means with the 97.5% credible interval are shown).**

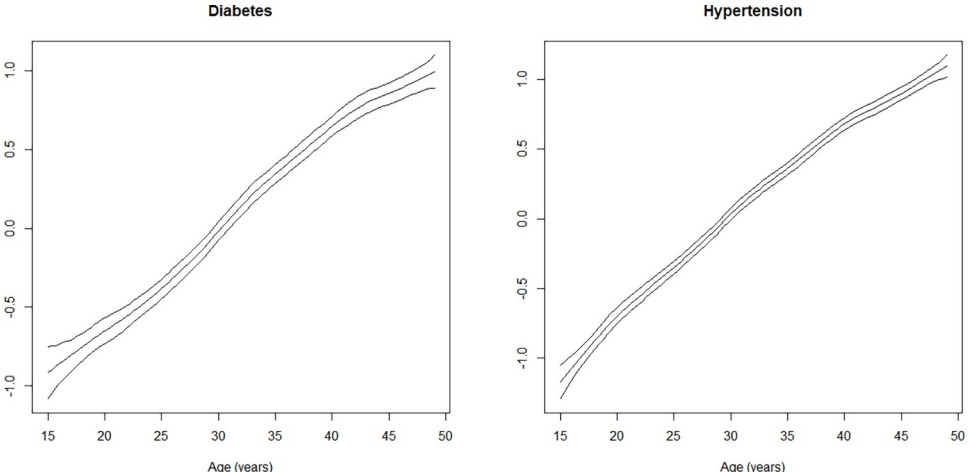

**Fig 4. Non-linear effects of age on the log-odds of diabetes and hypertension (posterior means along with the 97.5% credible interval are shown).**

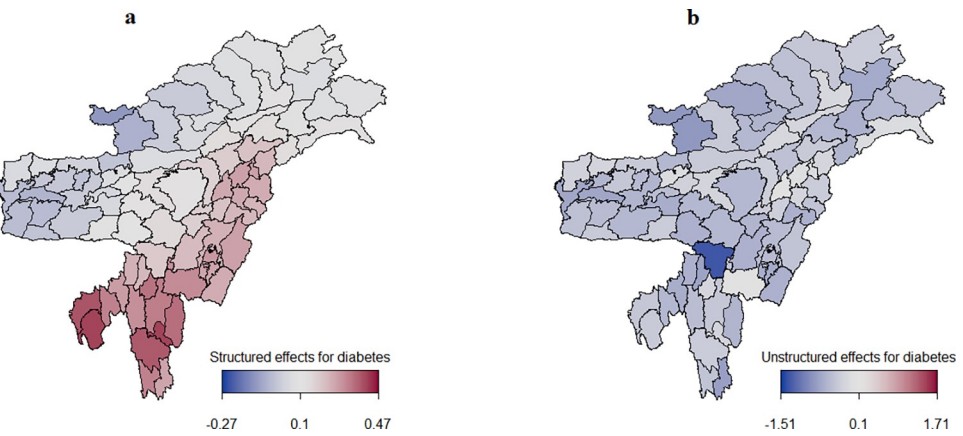

**Fig 5.** Estimated posterior means of the structured spatial effects (left) and the unstructured spatial effects (right) for the log-odds of diabetes.

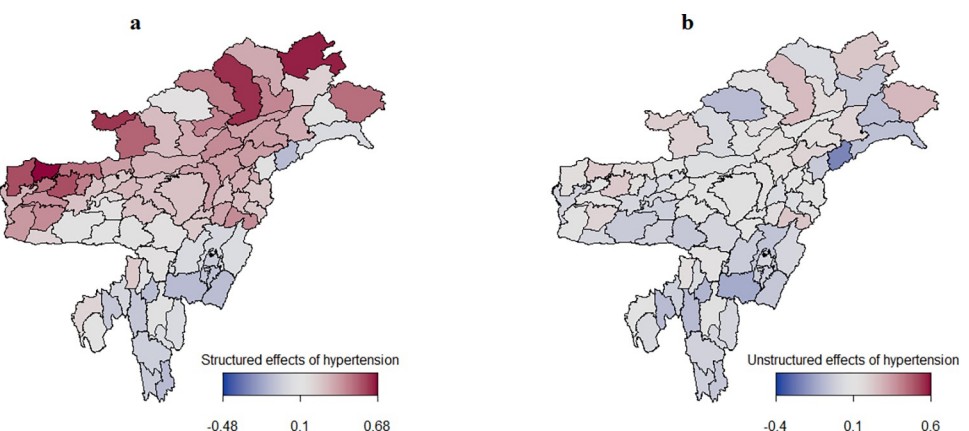

**Fig 6.** Estimated posterior means of the structured spatial effects (left) and the unstructured spatial effects (right) on the log-odds of hypertension.

## Discussion

This study attempted to explore the linear, non-linear, and spatial determinants of diabetes and hypertension among adults 15–49 years of age in Northeast India. The findings of this study reveal that the linear or fixed effect variables, namely sex of respondents, place of residence, marital status, and level of education, were significantly associated with the risk of diabetes and hypertension. Furthermore, the study noticed that the continuous variables, namely age of the respondents, body mass index, and household wealth index score, had a non-linear effect on the risk of diabetes and hypertension. This study adopted the geo-additive logistic approach to examine the relationship between diabetes and hypertension and their risk factors. The geo-additive model had the advantage of allowing mapping of the residual spatial effects to diabetes and hypertension while considering the effect of the non-linear covariates on the assumption of additivity.

In a geo-additive model, the spatial effect is the sum of the structured and unstructured spatial effects. This method has the advantage that it allows to account for possible unmeasurable factors and heterogeneity. In addition, the model allows for the exploration of the subtle influence of the non-linear relationship of the continuous covariates that is not possible in a linear model.

### Spatial effects for diabetes

The findings of the study reveal that the structured spatial effects for diabetes were relatively weaker in comparison with the unstructured spatial effects (Fig 5), meaning that the role of a district on the risk of diabetes was not similar to that of the neighbouring districts. This is an indication that geographical and environmental factors which surpass the boundaries of districts likely do not play any significant role in diabetes. With unstructured spatial effects for diabetes being dominant in this study, it can be concluded that there are unobserved district-specific influences that are not structured spatial effects (that is, not interrelated with those of neighboring districts) contributing to diabetes [38]. Such district-specific factors contributing to diabetes may include availability of healthcare facilities, cost and quality of healthcare, and cost of living. These factors may vary significantly between and within the states.

A study in Northeast India by Ngangbam & Roy found that people living in districts with many medical institutions and better road connectivity were more likely to seek formal healthcare services because of the easy accessibility [39]. Their study also revealed that high treatment

cost and poor quality of healthcare services reduced the probability of utilizing the healthcare services in a given place [39].

## Spatial effects for hypertension

The results indicate the clustering of hypertension in the districts of Arunachal Pradesh, Assam, and Nagaland (Fig 6). A high prevalence of hypertension in these three states has been reported in a previous study as well [40]. The present study revealed that structured spatial effects for hypertension dominated the unstructured spatial effects, meaning that the risk of hypertension in a particular district was similar to that in districts that were in close proximity. This is an indication that geographical and environmental factors surpassing district boundaries may have a significant role in hypertension. This clear structured spatial pattern for hypertension begs an explanation. The geographical or environmental factors contributing to hypertension may include lifestyle differences and urbanization [27]. One possible reason for the high prevalence of hypertension in Arunachal Pradesh may be that the region is located at a high altitude. A study in Tibet showed a strong correlation between the prevalence of hypertension and altitude, with every 100 m increase in altitude corresponding to a 2% increase in the prevalence of hypertension [41]. However, the relationship between hypertension and altitude is not clear and needs further investigation. Another possible explanation may lie in the intake of large amounts of sodium by way of salt that is added to yak butter tea. The consumption of yak butter tea helps to keep the body warm in the cold environment of the Himalayan mountains [42,43].

Studies suggest that consuming five cups or more of yak butter tea daily exposes an individual to a higher risk of hypertension as compared to those whose consumption is less [44]. Frequent consumption of salty butter tea may elevate the daily salt intake by four to five times, which is above the limits recommended by the World Health Organization [41]. But it is also well-known that even though people living at high altitudes are used to consuming large amounts of salt, they are less obese and fitter than those living at lower altitudes [45].

The high prevalence of hypertension in Arunachal Pradesh may also be attributed to the high alcohol consumption [46]. In Assam, it may be attributed to the high salt intake, higher body mass index, consumption of locally prepared alcohol, and central obesity [47]. In Nagaland, the high prevalence of hypertension may be attributed to lifestyle changes and changes in diet, which are direct outcomes of socioeconomic development and food consumption pattern [48].

## Fixed and non-linear effects

The fixed effect factors for diabetes and hypertension, which were significant in this study, were sex of the respondent, place of residence, marital status, and highest level of education (Tables 3 and 4). The influence of these factors on the risk of diabetes and hypertension is in agreement with the findings of previous studies [8,16]. The finding that men are more likely to suffer from diabetes and hypertension has also been reported in previous studies [49,50]. Men are associated with more smoking and a higher consumption of alcohol, both of which are common risk factors of diabetes and hypertension [5]. The results also demonstrate that the consumption of milk and eggs significantly reduces the risk of hypertension. An interesting finding was that the consumption of alcohol was associated with a lower risk of hypertension. One possible reason for this finding is that the current drinkers may have cut down on their alcohol intake to moderate levels, resulting in their blood pressure coming back to normal levels [51].

Body mass index (BMI) was found to have a non-linear relationship with diabetes and hypertension. The results of the non-linear effect of BMI reveal that the risk of diabetes and

hypertension was low among individuals having a normal BMI. It increased among those having BMI ranging from 30 to 40, then decreased among those with BMI ranging from 40 to 60, and then again increased among those having BMI above 60 (Fig 2). The non-linear effect of BMI on the risk of cardiovascular diseases and mortality has been reported in many studies [52,53]. Individuals with a higher BMI may not necessarily have high fat mass composition, but a high muscle or lean mass composition [54]. A higher amount of lean mass in an individual may act as a protective factor against cardiovascular disease and the individual may be considered healthy or having a good health [54,55]. By contrast, an individual may have a low BMI but a high body fat mass composition, increasing their likelihood of having cardiovascular disease [53].

Household wealth index score was found to have a non-linear relationship with diabetes and hypertension (Fig 3). The risk of diabetes was the highest among individuals with the richest wealth index score as compared to their counterparts having a poorer wealth index score. However, the risk of hypertension was the highest among individuals having the poorest wealth index score as compared to their counterparts having the richest wealth index score. Consistent with the previous studies, this study revealed that economic status is inversely related with the risk of hypertension [56–59]. Individuals with a higher income can afford to pay for a healthier lifestyle, including regular physical exercise and a healthier diet and benefit from accessibility to advanced and quality healthcare services. All such efforts likely reduce the risk of hypertension.

Our study is not without limitations. Firstly, it was cross-sectional in nature and, therefore, no causal inferences could be made from the results and findings. Secondly, since the study was based on secondary data sets, we were constrained to use only the variables found in the IDHS. Thirdly, the unavailability of district-level information on such factors as cost of living, cost of treatment for diabetes and hypertension, medical institutions, level of urbanization, availability of green space, and altitude meant that we were unable to ascertain the influence of these factors on the spatial variability of diabetes and hypertension. Despite the limitations, the strength of the study lies in the application of the Bayesian geo-additive model, which allowed for a joint estimation of fixed effect covariates, non-linear covariates, spatially structured variation, and spatially unstructured heterogeneity.

## Conclusion

In conclusion, it is evident that there are spatial effects for diabetes and hypertension in Northeast India. The results suggest that district-specific factors (that is, factors not related to neighboring districts) are most likely to increase the prevalence of diabetes. However, in the case of hypertension, factors found in districts in proximity to one another are most likely to increase its prevalence. Gender, place of residence, level of education, household wealth status, BMI, and consumption of egg and milk are significant to the risk of diabetes and hypertension. Besides considering the factors that are already known, diabetes and hypertension control measures for Northeast India should take into account the risk factors present within the districts and those related to the proximate districts as they possibly play a role in driving the spatial variability of diabetes and hypertension in the region. Evaluation of district-specific factors of diabetes within the region should be give importance.

## Supporting information

**S1 Table. Breakup of 82 districts by states in Northeast India.**
(PDF)

## Acknowledgments

The authors are grateful to the Demographic Health Survey (DHS) Program for providing the data for this study. Authors would like to thank the editor and two anonymous reviewers for their valuable comments and suggestions towards improvement of the paper. Also, the authors would like to thank Shailja Thakur for copyediting the manuscript.

## Author Contributions

**Conceptualization:** Strong P. Marbaniang, Holendro Singh Chungkham, Hemkhothang Lhungdim.

**Data curation:** Strong P. Marbaniang.

**Formal analysis:** Strong P. Marbaniang, Holendro Singh Chungkham.

**Investigation:** Strong P. Marbaniang.

**Methodology:** Strong P. Marbaniang, Holendro Singh Chungkham.

**Software:** Holendro Singh Chungkham.

**Supervision:** Hemkhothang Lhungdim.

**Writing – original draft:** Strong P. Marbaniang.

**Writing – review & editing:** Strong P. Marbaniang, Holendro Singh Chungkham, Hemkhothang Lhungdim.

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
