## [Decision Letter · Decision Letter 0]

27 Sep 2021

PONE-D-21-09955A Structured-additive modeling of Diabetes and Hypertension in Northeast IndiaPLOS ONE

Dear Dr. Marbaniang,

Thank you for submitting your manuscript to PLOS ONE. After careful consideration, we feel that it has merit but does not fully meet PLOS ONE’s publication criteria as it currently stands. Therefore, we invite you to submit a revised version of the manuscript that addresses the points raised during the review process.

We look forward to receiving your revised manuscript.

Kind regards,

Mohammad Asghari Jafarabadi

Academic Editor

PLOS ONE

Journal Requirements:

A clean copy of the edited manuscript (uploaded as the new *manuscript* file).

3. We note that Figures 1, 5 and 6 in your submission contain map images which may be copyrighted. All PLOS content is published under the Creative Commons Attribution License (CC BY 4.0), which means that the manuscript, images, and Supporting Information files will be freely available online, and any third party is permitted to access, download, copy, distribute, and use these materials in any way, even commercially, with proper attribution. For these reasons, we cannot publish previously copyrighted maps or satellite images created using proprietary data, such as Google software (Google Maps, Street View, and Earth). For more information, see our copyright guidelines: http://journals.plos.org/plosone/s/licenses-and-copyright.

a. You may seek permission from the original copyright holder of Figures 1, 5 and 6 to publish the content specifically under the CC BY 4.0 license.  

Reviewers' comments:

Reviewer's Responses to Questions

**Comments to the Author**

1. Is the manuscript technically sound, and do the data support the conclusions?

Reviewer #1: Yes

Reviewer #2: Yes

2. Has the statistical analysis been performed appropriately and rigorously? 

Reviewer #1: Yes

Reviewer #2: Yes

3. Have the authors made all data underlying the findings in their manuscript fully available?

Reviewer #1: Yes

Reviewer #2: Yes

4. Is the manuscript presented in an intelligible fashion and written in standard English?

Reviewer #1: Yes

Reviewer #2: Yes

5. Review Comments to the Author

Reviewer #1: I would like to mention the following comments:

1- In title and abstract, there is no refer to simulation.

2- The total number of database is not clear. Was sampling census (all people) or sampled? How?

3- Diabetes: The criteria for DM is not clear. Twice sampling with threshold of 126?

4- In addition to types of foods, the amount of consumption is also necessary.

5- There is no explanation about "Effect coding" in method section.

6- Table 1: It is not clear if this able is the results of real data or simulation data?

7- Table 2: M0, M1, M2 and M3 must be defined at the bottom of the table.

8- Why not random effects? Was non-linear effect fixed?

9- The interpretation of estimated compared to observed data is mising.

10- "Conclusions" subtitle might be better to change into "Conclusion".

Good Luck

Reviewer #2: The manuscript titled “A structured-additive modeling of diabetes and hypertension in Northeast India” addresses very important issue of diabetes and hypertension in India. The manuscript utilized the IDHS survey data and applied geo-additive logistic regression model to understand the influence of fixed effects and spatial heterogeneity. It also addresses the issue of non-linearity in spatial context. Overall, the work is important and study is conducted systematically to conclude about the findings.

There are some concerns which can be addressed before the manuscript can be accepted for publication

Abstract

L28: Rewrite the sentence as it is not clear.

L38: It can also be mentioned that why traditional linear regression models may fail to capture the spatial effects and why you choose Bayesian Geo-additive model. Very briefly, it can also be highlighted about the importance of accounting for spatial autocorrelation and how it can lead to bias in estimates.

L45, L46 and L47: The sentence can be re-written to make to clear about importance of unstructured effect for diabetes and structured effect for hypertension.

L49: It is not clear what you mean here by local and non-local factors.

L51: You mean to say here should “be” given more importance?

Introduction

L54 and L55: Reference can be added for these statements

L65-67: The statement can be re-written to make it clear

L67-69: Cite reference for this statement

L74: Sentence can be re-written a it appears you are mentioning about your findings in introduction.

L92: The reference cited (Ref No. 20) explores the availability of green spaces in neighborhood of individual households and it is not clear from your statement. It will be better to mention about the spatial scale here.

Materials and Methods

L112: Sentence is starting with abbreviation, kindly change it

L112: It can be mentioned in the reference when it was accessed/downloaded (Ref. No. 25)

L113: What necessary permissions were obtained? Please elaborate

L116: It can be made clear before this sentence that the survey done by IDHS and not done in this study to avoid confusion to readers

L122: There is some space in the total respondents which can be deleted – 112,062 and for 13,360

L127: The breakup of 82 districts sampled for each state can be mentioned.

L129: Rewrite the sentence to make it clear

L134: OMRON is trade name so can be named appropriately

L146: The explanatory variables were selected from the IDHS survey? If so, mention how many variables were there in the IDHS database and how many were selected in this study? In addition, there is no mention about the continuous variables (BMI, wealth index and age) in your methods, but you present the results.

L155: It is not justified why you choose to use multiple regression model for narrowing down to the number of variables. As mentioned in comment for L146, you need to mention the total number of variables. You have left to readers to count the variables. There are many variable selection methods to reduce the number of variables before applying your Bayesian Geo-additive model. You need to justify your method or you have to perform variable selection method before narrowing down to the number of variables

L158: Again, mention about the number of variables which were used for fitting the Geo-additive logistic regression model

L164: You need to mention how you are defining your neighbor here? Did you use adjacency matric or any other method to define neighborhood, then you need to mention about it and may write the equation/formula for the same.

L169: It is advisable to use two more models here to understand the influence of spatially structure and spatially unstructured heterogeneity. The two models can be combination of M1 and M2 with spatial structured heterogeneity in one model and spatial unstructured heterogeneity in the other model. In this way it can be known about the drop/increase in DIC.

L189: How it was arrived to use 40,000 iterations? The convergence of different parameters should be tested using Gelman-Rubin statistics and mentioned that how these values were. You have not performed any cross validation statistics using CPO and PIT which is required to know the model performance.

L196: It should also be mentioned how you are deciding significance of variables based on 97.5% credible interval and values which do not bridge zero were not considered significant

Results

L199: The overall prevalence can also be shown on a map so that it can be compared with the spatial structured and unstructured heterogeneity maps for diabetes and hypertension

L237, Table 3: If this is the result of your M3 model which you say is combination of M1 and M2 then why the co-efficient of your continuous variables are not shown in this table? You have separately shown the non-linear effect of continuous variable in figures but this should also be reflected in your table or justify why it was not done so. It is not clear from your methods that whether your non-linear model using continuous variables also included categorical variables. It should be mentioned clearly to avoid confusion to readers. Your final model has all the variables which is my concern as mentioned in comment for L155 you need to reduce the total number of variables for your final model so that significant variables can be rightly identified.

L249, Table 4: same comments as for L237

Discussion

L300: You can start your discussion by saying about significant variables in your final model and then about the capturing non-linear effect using Geo-additive model

L327: You can refer your figures in discussion.

L350: This point contradicts your findings as mentioned on L246, that individuals who consume alcohol are at lower risk of hypertension. If this is the case only for Arunachal Pradesh or other states?

L358: Refer table number of your results

L359: It can be re-written with proper citation

L375 & 376: You mention previous studies, but cite only one reference.

L376: Need to add reference for this statement

L381-L383: It can be justified why you were constrained to use only IDHS data and why other variables were not included. You mentioned in several places about other variables like medical institutions, health care facilities, cost of living, urbanization and altitude may be playing a role in driving your spatial variability. It will be important to mention why you did not include other variables as this would have helped in even planning interventions and resource allocation in high risk areas for diabetes and hypertension

Conclusion

L395: As pointed out previously you need to mention clearly what you mean by local and non-local factors and preferable use other words to describe this pattern.

6. PLOS authors have the option to publish the peer review history of their article (what does this mean?). If published, this will include your full peer review and any attached files.

Reviewer #1: **Yes: **Masoud Amiri

Reviewer #2: No

---

## [Author Response · Author response to Decision Letter 0]

1 Dec 2021

Reviewer #1: I would like to mention the following comments:

Comment 1- In title and abstract, there is no refer to simulation.

Response: Thank you for the comment. The present study uses the individual level data for the whole North-Eastern states of India from the latest round of Indian National Family Health Survey (INFHS). The model we applied is not on the simulated data. We hope this clarifies the point. 

Comment 2- The total number of database is not clear. Was sampling census (all people) or sampled? How?

Response: Thank you for the comment. The survey is not a census survey but it is a random sampling. We have incorporated a detailed explanation about the sampling procedure of the survey in the revised manuscript along with the reference. 

Comment 3- Diabetes: The criteria for DM is not clear. Twice sampling with threshold of 126?

Response: Thank you for the comment. The measuring of random blood glucose level was performed only once. The blood sample was collected at a random time during the day. Detailed explanation was incorporated in the revised manuscript.

Comment 4- In addition to types of foods, the amount of consumption is also necessary.

Response: Thank you for this important suggestion. We do agree with the reviewer that the quantity of food consumption is also an important covariate, however we could not incorporate this information in the analysis as this information is not available in the dataset. 

Comment 5- There is no explanation about "Effect coding" in method section.

Response: Thank you to the reviewer for giving an opportunity to clarify. The effect coding, we adopted is the zero-sum coding. The definition is mentioned in the heading of Table 1. As suggested we have included a short explanation in the manuscript about effect coding adopted on Page 8, L177. We adopted this coding just to compare the effects of categories in a particular covariate. The corresponding interpretations are in the Results section of the manuscript.

Comment 6- Table 1: It is not clear if this table is the results of real data or simulation data?

Response: Thank you for the comment. As mentioned earlier, figures provided in Table 1 are authors calculation from the real data that was used in the whole analysis (IDHS). 

Comment 7- Table 2: M0, M1, M2 and M3 must be defined at the bottom of the table.

Response: Thank you for highlighting this important point. We have incorporated the suggestion. 

Comment 8- Why not random effects? Was non-linear effect fixed?

Response: Thank you for the comment. The non-linear effect represents the influence of the continuous independent variables (e.g., age, BMI, wealth index score) on the dependent variable and the effect is not fixed but rather it allows for possible non-linear patterns in the model. 

Comment 9- The interpretation of estimated compared to observed data is missing.

Response: Thank you for the comment. The estimated values in Table 3 and Table 4 represent the posterior mean effect of the fixed covariates on the outcome variables. These values were estimated from the observed data using Model M3. Their interpretation has also been reported under the Fixed effect section.

Comment 10- "Conclusions" subtitle might be better to change into "Conclusion".

Response: Thank you for the comment. Now we have made the correction as suggested

Reviewer#2:

The manuscript titled “A structured-additive modeling of diabetes and hypertension in Northeast India” addresses very important issue of diabetes and hypertension in India. The manuscript utilized the IDHS survey data and applied geo-additive logistic regression model to understand the influence of fixed effects and spatial heterogeneity. It also addresses the issue of non-linearity in spatial context. Overall, the work is important and study is conducted systematically to conclude about the findings. 

Response: Thank you for appreciating our work. 

There are some concerns which can be addressed before the manuscript can be accepted for publication

Abstract

L28: Rewrite the sentence as it is not clear. 

Response: Thank you for the suggestion. We now have rephrased the sentences.

L38: It can also be mentioned that why traditional linear regression models may fail to capture the spatial effects and why you choose Bayesian Geo-additive model. Very briefly, it can also be highlighted about the importance of accounting for spatial autocorrelation and how it can lead to bias in estimates. 

Response: Thanks a lot to the reviewer for pointing out the issue. In order to make the readers more understandable regarding the model adopted we have added a short explanation of the advantage of geo-additive over traditional linear regression in the manuscript on Page 8, L186-189. We hope we clarify to the doubt and the reviewer is pleased.

L45, L46 and L47: The sentence can be re-written to make to clear about importance of unstructured effect for diabetes and structured effect for hypertension. 

Response: Thank you for the comment. We now have rephrased the sentences. 

L49: It is not clear what you mean here by local and non-local factors. 

Response: Thank you for the comment. By local factors we mean the unobserved risk factors present within the districts where the respondents reside. However, by non-local factors we mean the unobserved risk factors that are related to the nearby districts i.e., nearby districts may have similar risk factors

L51: You mean to say here should “be” given more importance?

Response: Thank you for point out this issue. We mean to say the same thing as per your suggestion and hence we have incorporated the suggestion.

Introduction

L54 and L55: Reference can be added for these statements

Response: Comment incorporated.

L65-67: The statement can be re-written to make it clear

Response: Comment incorporated

L67-69: Cite reference for this statement

Response: Thank you for the suggestion. We now have incorporated the reference

L74: Sentence can be re-written a it appears you are mentioning about your findings in introduction. 

Response: Comment incorporated. 

L92: The reference cited (Ref No. 20) explores the availability of green spaces in neighborhood of individual households and it is not clear from your statement. It will be better to mention about the spatial scale here. 

Response: Thank you for the suggestion. We now have incorporated the suggestion. 

Materials and Methods

L112: Sentence is starting with abbreviation, kindly change it

Response: We now have made the necessary changes

L112: It can be mentioned in the reference when it was accessed/downloaded (Ref. No. 25)

Response: Accessed date is now incorporated in the reference.

L113: What necessary permissions were obtained? Please elaborate

Response: Before getting access to download the DHS dataset, it is compulsory for the data user to register on the website https://dhsprogram.com/data/new-user-registration.cfm. After the registration is complete the user will receive an email notification in the registered email id which will give the permission to download the data. 

L116: It can be made clear before this sentence that the survey done by IDHS and not done in this study to avoid confusion to readers

Response: Thank you for the suggestion. We now have elaborated in the revised manuscript that IDHS was conducted by the International Institute for Population Sciences (IIPS), Mumbai a nodal agency appointed by the Ministry of Health and Family Welfare, Government of India. 

L122: There is some space in the total respondents which can be deleted – 112,062 and for 13,360

Response: Thank you for highlighting this error. We now have made the correction.

L127: The breakup of 82 districts sampled for each state can be mentioned.

Response: Table of breakup of 82 districts by States was attached as supplementary information in the revised manuscript. 

L129: We now have rewritten the sentence to make it clear

Response: Thank you for the comment. Now the sentences were rephrased.

L134: OMRON is trade name so can be named appropriately

Response: Thank you for the suggestion. We now have renamed appropriately.

L146: The explanatory variables were selected from the IDHS survey? If so, mention how many variables were there in the IDHS database and how many were selected in this study? In addition, there is no mention about the continuous variables (BMI, wealth index and age) in your methods, but you present the results. 

Response: The IDHS questionnaire obtained information on Household characteristics, Women and Child characteristics, Men, and Biomarker. 

Household dataset have 5183 variables.

Women and Child dataset have 4797 variables

Men dataset have 747 variables

Biomarker dataset have 270 variables

As guided by the literature review, we have selected 21 explanatory variables for this study namely Age, Gender, Place of residence, Marital status, Level of education, Wealth Index Score, Body Mass Index, Caste, Consumption of milk, pulses, vegetables, fruits, egg, fish, chicken, fried food, aerated drinks, alcohol, cigarette, tobacco. 

We now have included the explanatory variables BMI, Age, and wealth Index score in the methods section. 

L155: It is not justified why you choose to use multiple regression model for narrowing down to the number of variables. As mentioned in comment for L146, you need to mention the total number of variables. You have left to readers to count the variables. There are many variable selection methods to reduce the number of variables before applying your Bayesian Geo-additive model. You need to justify your method or you have to perform variable selection method before narrowing down to the number of variables

Response: Thanks a lot to the reviewer. Actually, the selection of variables are guided by review of literature and available data in the context of India. First, we included more plausible variables and apply the usual linear regression model to select variables with a significance level of 20% in order to keep more plausible variables before applying the complex and time consuming geo-additive models. We do agree that the selection can be done in the geo-additive models itself. But because of time consumed in running one model for geo-additive, we did not adopt this approach. We hope the reviewer is satisfied with the response.

L158: Again, mention about the number of variables which were used for fitting the Geo-additive logistic regression model

Response: Thanks a lot. We have mentioned the variables retained for the geo-additive model on Page 8, L185.

L164: You need to mention how you are defining your neighbor here? Did you use adjacency matric or any other method to define neighborhood, then you need to mention about it and may write the equation/formula for the same.

Response: Thanks a lot to the reviewer for the suggestion. Instead of writing a complex equation, we have included few lines about the construction of neighbours on Page 10, L216-221.

L169: It is advisable to use two more models here to understand the influence of spatially structure and spatially unstructured heterogeneity. The two models can be combination of M1 and M2 with spatial structured heterogeneity in one model and spatial unstructured heterogeneity in the other model. In this way it can be known about the drop/increase in DIC.

Response: Thanks a lot for the comment. We would like to clarify that in the old manuscript we have mistakenly written the AIC value instead of the Deviance value in Table 2. Now, in the revised manuscript we have correctly written the Deviance value (Table 2). We hope the reviewer accept our mistake. 

As suggested we have included two more models i.e. M3 and M4. Then M3 is re-named as M5 as the best model. We considered the change in DIC and used the difference of less than 3 as negligible (Besag J, Kooperberg C: On conditional and intrinsic autoregressions. Biometrika 1995, 82:733–746). We found the differnces in DIC as: M0-M5=2941.24, M1-M5=6890.98, M2-M5=469.08, M3-M5=7.62, M4-M5=3.35 for Diabetes, which are all greater than 3. Similarly, for hypertension the differences are 7402.04, 16242.97, 991.37, 4.76, 4.20 respectively. We hope the reviewer is satisfied.

L189: How it was arrived to use 40,000 iterations? The convergence of different parameters should be tested using Gelman-Rubin statistics and mentioned that how these values were. You have not performed any cross-validation statistics using CPO and PIT which is required to know the model performance.

Response: Thanks a lot for pointing out this issue. In order to check we set enough number of iterations while running the model. In this way we set the iterations as 40,000. We do understand it is also important to check the predictive power of the fitted model. We however would like to emphasize that we have checked the convergence through trace plots for all the estimated parameters. We found no autocorrelations in the trace plots. We are very sorry to say that the package we used “bamlss” does not support to calculate/plot the PIT additional out of sample measures. We, however checked whether any of the cpo values are non-zero. We found none of them are non-zero for both the outcomes i.e. hypertension and diabetes. As suggested by literatures we checked the negative of the mean of logarithm of cpo values for the final fitted model i.e. M5. We found the values as: 0.4012 for hypertension and 0.2189. We hope these are some of the measures that can be provided in addition to the measures DIC, pD which are popular measures in Bayesian analysis. Regarding the Gelman-Rubin statistics, we need to run the final model for at least 2 chains which takes a lot of time and need to present many figures for that many variables. Just to make a clarification to the reviewer we provide some figures along with gelman plot. We would like to emphasize that the command we used for Gelman-Rubin diagnostics provided by the package “coda” is that it might mis-diagnose convergence if the shrink factor happens to be close to 1 by chance. Looking at the values provided by the Gelman-Rubin statistics, none of them are extremely well above the value 1, this confirms the convergence as we found in the trace plots.

Diabetes model:

Potential scale reduction factors:

 Point est. Upper C.I.

pi.s.s(age).b1 1.02 1.09

pi.s.s(age).b2 1.03 1.12

pi.s.s(age).b3 1.01 1.06

pi.s.s(age).b4 1.01 1.04

pi.s.s(age).b5 1.00 1.00

pi.s.s(age).b6 1.01 1.03

pi.s.s(age).b7 1.01 1.05

pi.s.s(age).b8 1.01 1.04

pi.s.s(age).b9 1.00 1.02

pi.s.s(age).b10 1.01 1.03

pi.s.s(age).b11 1.02 1.08

pi.s.s(age).b12 1.02 1.10

pi.s.s(age).b13 1.02 1.10

pi.s.s(age).b14 1.01 1.05

pi.s.s(age).b15 1.01 1.04

pi.s.s(age).b16 1.02 1.10

pi.s.s(age).b17 1.03 1.15

pi.s.s(age).b18 1.01 1.05

pi.s.s(age).b19 1.00 1.00

pi.s.s(age).tau21 1.01 1.04

Multivariate psrf

1.05

Hypertension model:

Potential scale reduction factors:

 Point est. Upper C.I.

pi.s.s(age).b1 1.004 1.017

pi.s.s(age).b2 0.999 1.000 

pi.s.s(age).b3 1.001 1.007

pi.s.s(age).b4 1.000 1.003

pi.s.s(age).b5 1.003 1.007

pi.s.s(age).b6 1.000 1.002

pi.s.s(age).b7 1.000 1.000

pi.s.s(age).b8 1.000 1.001

pi.s.s(age).b9 1.003 1.012

pi.s.s(age).b10 0.999 1.000

pi.s.s(age).b11 0.999 0.999

pi.s.s(age).b12 1.000 1.001

pi.s.s(age).b13 1.001 1.002

pi.s.s(age).b14 1.003 1.005

pi.s.s(age).b15 1.002 1.014

pi.s.s(age).b16 1.006 1.020

pi.s.s(age).b17 1.003 1.017

pi.s.s(age).b18 1.001 1.003

pi.s.s(age).b19 1.000 1.001

pi.s.s(age).tau21 1.062 1.088

Multivariate psrf

1.02

In the same way we can check for other terms like “bmi” and “wealth index”. We hope these clarify to the points raised by the reviewer to some extend.

L196: It should also be mentioned how you are deciding significance of variables based on 97.5% credible interval and values which do not bridge zero were not considered significant

Response: Thanks a lot for giving us to make a clarification. Actually, there is no as such theoretical reason behind using 97.5%. We also adopted the generally adopted significance level. Just would like to mention that these are actually the 95% credible intervals. 

Results

L199: The overall prevalence can also be shown on a map so that it can be compared with the spatial structured and unstructured heterogeneity maps for diabetes and hypertension

Response: Thanks a lot for the comment. Yes, we do agree the reviewer’s view. But because of small numbers of districts in the map, it will be very messy to show all the prevalences on the map. Therefore, we dropped the idea to show on the map. 

L237, Table 3: If this is the result of your M3 model which you say is combination of M1 and M2 then why the co-efficient of your continuous variables are not shown in this table? You have separately shown the non-linear effect of continuous variable in figures but this should also be reflected in your table or justify why it was not done so. It is not clear from your methods that whether your non-linear model using continuous variables also included categorical variables. It should be mentioned clearly to avoid confusion to readers. Your final model has all the variables which is my concern as mentioned in comment for L155 you need to reduce the total number of variables for your final model so that significant variables can be rightly identified. 

Response: Thanks a lot for the comment. Basically, we wanted to see the whether there are non-linear effects of age, bmi and wealth index in the model and did not intend to understand the effects as such. We therefore thought that it is better to show as a figure as done in most of studies. Yes, the combined model includes both fixed and non-linear effects of the variables age, BMI and wealth index. To make it clearer to the readers we have elaborated

L249, Table 4: same comments as for L237

Response: Thanks a lot for the comments. Now we have changed completely on Pages 9 and 10, L198-225.

Discussion

L300: You can start your discussion by saying about significant variables in your final model and then about the capturing non-linear effect using Geo-additive model

Response: Thank you for the suggestion. We now have incorporated the suggestion. 

L327: You can refer your figures in discussion. 

Response: Thank you for the suggestion. Now figure reference have been incorporated in the discussion. 

L350: This point contradicts your findings as mentioned on L246, that individuals who consume alcohol are at lower risk of hypertension. If this is the case only for Arunachal Pradesh or other states?

Response: Thank you for the comment. The statement in L350 we mean to explain for Arunachal Pradesh only, as it is well known that Arunachal Pradesh had the highest consumption of alcohol in India and high prevalence of hypertension. However, the statement in line L246 explains for the whole of Northeast India. 

L358: Refer table number of your results

Response: Comment Incorporated

L359: It can be re-written with proper citation 

Response: Thank you for the comment. Now the sentences have been re-written with proper citation. 

L375 & 376: You mention previous studies, but cite only one reference. 

Response: Thank you for the comment. We now have added some reference from the related articles. 

L376: Need to add reference for this statement 

Response: This sentence has been rephrased with proper references. 

L381-L383: It can be justified why you were constrained to use only IDHS data and why other variables were not included. You mentioned in several places about other variables like medical institutions, health care facilities, cost of living, urbanization and altitude may be playing a role in driving your spatial variability. It will be important to mention why you did not include other variables as this would have helped in even planning interventions and resource allocation in high-risk areas for diabetes and hypertension 

Response: Thank you for the important suggestion. I do agree with you that medical institutions and healthcare facilities, cost of living index, urbanization, and altitude may play a role in driving the spatial clustering of diabetes and hypertension. Since these information’s was not available in the IDHS data, hence we cannot study their influence on the spatial variability of diabetes and hypertension.

 Considering your suggestion, we have included this as one of the limitations in our study. 

Conclusion

L395: As pointed out previously you need to mention clearly what you mean by local and non-local factors and preferable use other words to describe this pattern. 

Response: Thank you for the suggestion. By local factors we mean the risk factors present within the districts and non-local factors mean the risk factors which are related to proximate districts. We have renamed and rephrased the sentences.

---

## [Decision Letter · Decision Letter 1]

30 Dec 2021

A Structured Additive Modeling of Diabetes and Hypertension in Northeast India

PONE-D-21-09955R1

Dear Dr. Marbaniang,

We’re pleased to inform you that your manuscript has been judged scientifically suitable for publication and will be formally accepted for publication once it meets all outstanding technical requirements.

Kind regards,

Mohammad Asghari Jafarabadi

Academic Editor

PLOS ONE

Additional Editor Comments (optional):

Reviewers' comments:

Reviewer's Responses to Questions

**Comments to the Author**

1. If the authors have adequately addressed your comments raised in a previous round of review and you feel that this manuscript is now acceptable for publication, you may indicate that here to bypass the “Comments to the Author” section, enter your conflict of interest statement in the “Confidential to Editor” section, and submit your "Accept" recommendation.

Reviewer #1: All comments have been addressed

Reviewer #2: All comments have been addressed

2. Is the manuscript technically sound, and do the data support the conclusions?

Reviewer #1: Yes

Reviewer #2: Yes

3. Has the statistical analysis been performed appropriately and rigorously? 

Reviewer #1: Yes

Reviewer #2: Yes

4. Have the authors made all data underlying the findings in their manuscript fully available?

Reviewer #1: Yes

Reviewer #2: Yes

5. Is the manuscript presented in an intelligible fashion and written in standard English?

Reviewer #1: Yes

Reviewer #2: Yes

6. Review Comments to the Author

Reviewer #1: In answer to reviewers and revised main text, all comments have been considered and appropriately addressed.

Reviewer #2: Thank you for addressing all the issues raised. Overall the manuscript is important contribution towards understanding the spatial variability in diabetes and hypertension in the region by using robust Bayesian Geo-additive model. Overall good piece of work.

7. PLOS authors have the option to publish the peer review history of their article (what does this mean?). If published, this will include your full peer review and any attached files.

Reviewer #1: **Yes: **Masoud Amiri

Reviewer #2: No

---

## [Editor Report · Acceptance letter]

5 Jan 2022

PONE-D-21-09955R1 

A Structured Additive Modeling of Diabetes and Hypertension in Northeast India 

Dear Dr. Marbaniang:

I'm pleased to inform you that your manuscript has been deemed suitable for publication in PLOS ONE. Congratulations! Your manuscript is now with our production department. 

Kind regards, 

on behalf of

Professor Mohammad Asghari Jafarabadi 

Academic Editor

PLOS ONE